# Clinical Significance of Systemic Inflammation Markers in Newly Diagnosed, Previously Untreated Hepatocellular Carcinoma

**DOI:** 10.3390/cancers12051300

**Published:** 2020-05-21

**Authors:** Jeong Il Yu, Hee Chul Park, Gyu Sang Yoo, Seung Woon Paik, Moon Seok Choi, Hye-Seung Kim, Insuk Sohn, Heerim Nam

**Affiliations:** 1Department of Radiation Oncology, Samsung Medical Center, Sungkyunkwan University School of Medicine, 81 Irwon-ro, Gangnam-gu, Seoul 06351, Korea; ro.yuji651@samsung.com (J.I.Y.); gs.levin.yoo@samsung.com (G.S.Y.); 2Department of Medical Device Management and Research, Samsung Advanced Institute for Health Sciences and Technology, Sungkyunkwan University, 81 Irwon-ro, Gangnam-gu, Seoul 06351, Korea; 3Departments of Medicine, Samsung Medical Center, Sungkyunkwan University School of Medicine, 81 Irwon-ro, Gangnam-gu, Seoul 06351, Korea; sw.paik@samsung.com (S.W.P.); drms.choi@samsung.com (M.S.C.); 4Statistics and Data Center, Samsung Medical Center, 81 Irwon-ro, Gangnam-gu, Seoul 06351, Korea; hyeseung.kim@sbri.co.kr (H.-S.K.); insuks@gmail.com (I.S.); 5Department of Radiation Oncology, Gangbook Samsung Hospital, Sungkyunkwan University School of Medicine, 29, Saemunan-ro, Jongno-gu, Seoul 03181, Korea; heerim.nam@samsung.com

**Keywords:** inflammation, liver cancer, prognostic factor, registry, survival, neutrophil, lymphocyte, monocyte, platelet, ratio

## Abstract

This study aimed to investigate the clinical significance of systemic inflammation markers (SIMs)—including neutrophil-to-lymphocyte ratio (NLR), platelet-to-lymphocyte ratio (PLR), and lymphocyte-to-monocyte ratio (LMR)—in patients with newly diagnosed, previously untreated hepatocellular carcinoma (HCC). The present study was performed using prospectively collected registry data of newly diagnosed, previously untreated HCC from a single institution. The training set included 6619 patients from 2005 to 2013 and the validation set included 2084 patients from 2014 to 2016. The SIMs as continuous variables significantly affected the overall survival (OS), and the optimal cut-off value of NLR, PLR, and LMR was 3.0, 100.0, and 3.0, respectively. There were significant correlations between SIMs and the albumin-bilirubin grade/Child-Turcotte-Pugh class (indicative of liver function status) and the staging system/portal vein invasion (indicative of the tumor burden). The OS curves were well stratified according to the prognostic model of SIMs and validated using the bootstrap method (1000 times, C-index 0.6367, 95% confidence interval (CI) 0.6274–0.6459) and validation cohort (C-index 0.6810, 95% CI 0.6570–0.7049). SIMs showed significant prognostic ability for OS, independent of liver function and tumor extent, although these factors were significantly correlated with SIMs in patients with newly diagnosed, previously untreated HCC.

## 1. Introduction

In liver cancer patients, hepatocellular carcinoma (HCC) accounts for approximately 90%, which remains the second-leading cause of cancer ranked by absolute years of life lost worldwide in men [1], despite the remarkable progress in the background knowledge about baseline liver disease and the management of HCC. Although the TNM classification system of the American Joint Committee on Cancer (AJCC)/International Union Against Cancer (UICC) provides information about the classification of HCC [2], the system is not widely used because of its limited ability in determining the optimal treatment modalities as well as the prognostic factors, unlike its use for other malignancies [3]. Because most cases of HCC develop in patients with chronic hepatitis with or without liver cirrhosis, it is crucial to not only manage HCC in those patients according to the TNM staging system, but also maintain liver function and treat or suppress reactivation of baseline hepatitis.

Traditionally, the Child-Turcotte-Pugh classification has been used to assess the prognosis of liver function in patients with chronic liver disease, including liver cirrhosis, which is the most important prognostic factor of HCC [4]. Recently, the albumin-bilirubin (ALBI) grade, which is a simpler and easier tool for evaluating liver function than the Child-Turcotte-Pugh classification, showed non-inferior outcomes in a large-scale validation study; therefore, ALBI grade is also frequently widely used for evaluating the baseline liver function of patients with HCC [5]. 

Although the ALBI grade and Child-Turcotte-Pugh classification predict the prognosis of HCC very well by evaluating baseline liver function, they do not accurately reflect the condition and extent of the tumor. Accordingly, several systems are being suggested to overcome these limitations, and the tumor status, including the TNM stage, was mainly used to modify the ALBI grade [6]. 

The clinical significance of systemic inflammation—assessed with several laboratory panels of blood—has been recently validated for various cancers [7,8,9]. In fact, the neutrophil-to-lymphocyte ratio (NLR), platelet-to-lymphocyte ratio (PLR), and/or lymphocyte-to-monocyte ratio (LMR) showed positive ability as prognostic and/or predictive factors after several treatments for HCC as well as other malignancies [7,8,9,10,11]. In particular, NLR is a strong predictive factor of immune checkpoint blockade, which has been replacing conventional standard treatment for many cancers, including HCC [9,10,12].

Using this background, we conducted the present study to evaluate the clinical significance of systemic inflammation markers (SIMs), including NLR, PLR, and LMR, in patients with newly diagnosed, previously untreated HCC.

## 2. Results

### 2.1. Patients

Table 1 shows the baseline characteristics of the patients in the training set (6619 HCC patients registered from January 2005 to December 2013) and validation set (2084 HCC patients registered from January 2014 to December 2016) in the prospective HCC registry. 

There were significant differences in several characteristics between both of the sets. In particular, Child-Turcotte-Pugh class A and ALBI grade I, both of which indicated good liver function status, were significantly higher in the validation set. Considering the primary treatment, liver resection was increasingly performed in the validation set, while liver transplantation decreased slightly. 

### 2.2. Prognostic Significance and Optimal Cut-Off Values of Systemic Inflammation Markers for OS

Table 2 shows the univariate analysis outcomes of overall survival (OS) according to the probable prognostic factors, including SIMs, for the 6619 patients in the training set. OS was significantly associated with known prognostic factors, including ALBI grade, alpha-fetoprotein (AFP), Eastern Cooperative Oncology Group (ECOG) performance status, portal vein invasion (PVI), and TNM stage. In addition, the continuous variables of NLR (*p <* 0.0001, hazard ratio (HR) 1.082, 95% confidence interval (CI) 1.073–1.092), PLR (*p <* 0.0001, HR 1.005, 95% CI 1.004–1.005), and LMR (*p <* 0.0001, HR 0.811, 95% CI 0.795–0.827) were also significantly associated with OS.

To determine the optimal cut-off value of NLR, PLR, and LMR for OS, Cox regression analysis was performed for each ratio. The C-index and 95% CI for each ratio are shown in Appendix A. The maximum C-index of NLR, PLR, and LMR was observed at the point of 2.2 (C-index 0.609, 95% CI 0.600–0.618), 97.5 (C-index 0.589, 95% CI 0.581–0.598), and 3.7 (C-index 0.611, 95% CI 0.602–0.619), respectively. The final cut-offs in the present study were 3.0 for NLR (C-index 0.594, 95% CI 0.586–0.602), 100.0 for PLR (C-index 0.589, 95% CI 0.580–0.597), and 3.0 for LMR (C-index 0.601, 95% CI 0.602–0.619), all of which were close to the point of the maximum C-index in the present study and repeatedly verified in other studies. According to the cut-off values, when the NLR was ≥3, the PLR was ≥100 and the LMR was ≤3—patients with these values were classified into the risk group. Kaplan–Meier curves of OS according to the SIMs as categorical variables are displayed in Appendix A. 

### 2.3. Correlation Analysis between SIMs and Other Prognostic Factors

As continuous variables, NLR (correlation coefficient 0.0977, *p <* 0.001), PLR (correlation coefficient −0.0830, *p <* 0.001), and LMR (correlation coefficient −0.2432, *p <* 0.001) were significantly associated with the ALBI grade. Moreover, the NLR (*p <* 0.001), PLR (*p =* 0.028), and LMR (*p <* 0.001) as categorical variables with cut-off values of 3.0, 100.0, and 3.0 respectively, also showed significant correlations with the ALBI grade. The correlation between Child-Turcotte-Pugh class and the PLR was not significant, in contrast to the correlation shown between Child-Turcotte-Pugh class and NLR and LMR. There was also a significant correlation between those markers and the factors associated with tumor burden, including stage (Barcelona Clinic Liver Cancer (BCLC) and AJCC/UICC TNM), portal vein invasion (PVI), and tumor markers. Glasgow Prognostic Score (GPS) can be analyzed only in patients who selectively had C-reactive protein (CRP) measured [13] and showed a significant association with all SIMs. The results of correlation analyses between SIMs and variables are displayed in Appendix A.

### 2.4. Prognostic Significance of SIMs on OS in Multivariate Analysis

Table 3 shows the outcomes of multivariate analysis performed with SIMs as categorical variables and the significant prognostic factors on univariate analysis. The NLR (*p <* 0.0001, HR 1.363, 95% CI 1.238–1.501), PLR (*p <* 0.0001, HR 1.465, 95% CI 1.349–1.591), and LMR (*p =* 0.0006, HR 1.168, 95% CI 1.068–1.277) as categorical variables also showed significant prognostic ability on multivariate analysis. 

### 2.5. Clinical Significance of SIMs in the Validation Set 

On univariate analysis, the prognostic significance of SIMs on OS was successfully validated in the cohort registered from January 2014 to December 2016, as displayed in Appendix A. In addition, on multivariate analysis, NLR (*p <* 0.001, HR 3.272, 95% CI 2.712–3.948), PLR (*p <* 0.001, HR 2.477, 95% CI 2.063–2.974), and LMR (*p <* 0.001, HR 3.290, 95% CI 2.746–3.943) as categorical variables showed significant prognostic ability for OS. 

There were significant correlations between SIMs (NLR and LMR) and ALBI grade (both *p <* 0.001), except PLR (*p =* 0.259). Moreover, PLR was not significantly correlated with Child-Turcotte-Pugh class, in contrast to NLR and LMR. However, on correlation analysis between SIMs and other variables, including stage (BCLC and AJCC/UICC TNM), PVI, and tumor markers, significant correlation was observed for all three markers with the findings of the training set. Hepatic steatosis index (HSI) was only available to analyze in the validation set [14], and there was a significant correlation between HSI and LMR. GPS was also only available in patients who selectively had CRP measured, and showed a significant association with all SIMs. The results of correlation analysis between SIMs as categorical variables and other variables are displayed in Appendix A.

### 2.6. OS According to the Prognostic Model Based on SIMs in the Training and Validation Sets

The prognostic model established based on the number of SIMs was used to evaluate OS in the training and validation sets, and, OS was also successfully validated internally using the bootstrap method (1000 times, C-index 0.6367, 95% CI 0.6274–0.6459) and externally (C-index 0.6810, 95% CI 0.6570–0.7049). The OS curves were well stratified in the training and validation sets among four risk groups classified by the number of SIMs (Figure 1). 

There were differences in the prognosis of each set according to the prognostic model based on SIMs, considering the treatment policy and ALBI grade in both the training and validation sets. The OS curves were well separated according to the SIMs in the subgroup treated with only palliative or supportive care in both the training and validation sets (both *p <* 0.001). The difference in OS curves was not distinct in patients treated with curative intent, including liver transplantation, liver resection, or radiofrequency ablation (RFA). The SIMs could stratify the OS curves well in the subgroup with ALBI grade 1 and 2 in both of the sets (all *p <* 0.001). In contrast, there was no difference in the curves of the subgroup treated with curative intent and ALBI grade 3 in both the training and validation sets. Kaplan–Meier OS curves according to the SIMs and treatment policy or ALBI grade sets are displayed in Figure 2 (training set) and Figure 3 (validation set).

There were also significant differences in the prognosis of each set according to the SIMs according to the BCLC staging system and treatment modalities. The OS curves were best distinguished according to the SIMs in the BCLC C subgroup and in the subgroup treated with trans-arterial chemoembolization (TACE) in both sets, as displayed in Appendix A. 

## 3. Discussion

In the present study, that evaluated the clinical significance of SIMs including NLR, PLR, and LMR in newly diagnosed, previously untreated HCC, the SIMs showed significant correlation with previously recognized prognostic factors, including ALBI grade and Child-Turcotte-Pugh class (representing baseline liver function) and TNM stage and PVI (representing tumor burden). These markers showed significant prognostic ability, independent of the previously recognized prognostic factors of HCC. The SIMs, in particular, showed a higher prognostic performance for cases of advanced HCC and/or those managed with palliative care. This result might provide additional useful information regarding the pre-existing prognostic models to determine appropriate management and prognosticators in these patients. 

Neutrophils are the first-line defensive cells in the innate arm of the immune system. In the liver, neutrophils as the mediators of inflammation can be induced to express several mediators that can influence inflammatory and immune responses, thereby affecting liver injury [15]. The reduction in the neutrophil count during hepatitis C treatment with peginterferon was associated with achieving a sustained virological response [16]. In the present study, NLR as a continuous as well as a categorical variable was significantly correlated with ALBI grade and Child-Turcotte-Pugh class in newly diagnosed HCC. This finding possibly supports the hypothesis that liver damage caused by neutrophils is associated with liver function deterioration [15].

It is well known that neutrophils exacerbate, rather than suppress, cancer progression by promoting the initiation, growth, proliferation, and angiogenesis, as well as suppression of senescence and antitumor immunity [16,17]. The functions of neutrophil induction in the early steps of the metastatic cascade include intravasation and formation of the premetastatic niche [18,19]. Clinical evidence shows that NLR supports the above-mentioned preclinical findings that neutrophils promote cancer progression, showing its prognostic significance in recurrence as well as survival in various cancers, including HCC [9,12]. Even in the present study, NLR was significantly associated with the variables that represented tumor burden in HCC, such as TNM/BCLC stage and AFP/protein induced by vitamin K absence or antagonist-II (PIVKA-II), although the causal relationship between them could not be determined. NLR was one of the significant prognostic factors of OS on multivariate analysis. However, the prognostic significance of NLR was more prominent in the patients who received palliative treatment, such as TACE, radiotherapy, sorafenib, or supportive care, in both the training and validation sets. OS was not different according to NLR, especially in patients who underwent liver transplantation, and the difference was very limited in patients treated with liver resection or RFA.

Platelets—multipurpose cytoplasmic fragments of megakaryocytes—are essential for hemostasis/thrombosis, vascular integrity, angiogenesis, wound healing, inflammation/immune and liver regeneration, and platelet counts are frequently reduced in patients with chronic liver disease [20,21]. In contrast, tumor cells activate and aggregate platelets, which in turn can sustain proliferative signals, support cancer stem cells, prevent cancer death, induce angiogenesis, and cause metastasis while evading immune detection [20]. A high platelet count and/or platelet volume is associated with poor prognosis in various cancers [22]. High platelet counts are independent prognostic factors of HCC after liver resection [21]. In contrast, a low platelet count was significantly associated with a lower OS and recurrent-free survival after liver resection [23]. In the present study, there was a significant correlation between PLR and other well-recognized prognostic factor of HCC, including TNM/BCLC stage and AFP/protein induced by vitamin K absence or antagonist-II (PIVKA-II). Moreover, PLR was a significant and independent prognostic factor of OS on multivariate analysis. Similar to the results of NLR, the prognostic significance of PLR was lower in patients who received curative treatment, including liver transplantation, liver resection, or RFA. There was a negative correlation between PLR and ALBI grade, although both factors were significant prognostic factors of OS. This phenomenon may reflect the effect of platelets on chronic liver decrease or liver regeneration as well as on tumor activation and exacerbation.

Because of the complex effect of platelets on inflammation and tumor progression, as well as on chronic liver disease and liver regeneration, the prognostic role of platelets in HCC should be further investigated. 

Lymphocytes are the backbone of the immune system, which is the pivotal defense mechanism of the microenvironment in which HCC develops on a background of chronic inflammation owing to hepatitis B/C virus, toxic agents including alcohol, or metabolic syndromes [24,25]. The prognostic significance of tumor-infiltrating lymphocytes is well recognized [26]. Serum lymphocytes are also one of the significant prognosticators as a part of NLR as well as the absolute lymphocyte count in HCC [27]. Moreover, elevated serum monocyte counts are associated with HCC progression [28]. In the present study, LMR as a continuous variable showed the highest correlation between ALBI grade and SIMs: as the LMR kept increasing, the ALBI grade kept decreasing. LMR is also significantly correlated with well-recognized prognostic factors of HCC, including the staging systems, tumor markers, and independent prognostic factor of OS. 

Our findings suggest that SIMs, including NLR, PLR, and LMR, are not only independent prognostic factors for OS but are also closely associated with the tumor extent and liver function in newly diagnosed, previously untreated HCC. In particular, PLR showed conflicting outcomes in terms of the OS and liver function. With the notable achievement in the field of immune-oncology, there is a growing interest and need for valuable biomarkers encompassing tumor microenvironment, including immunity. As a prognosticator, the Immunoscore based on the immune cells that infiltrate cancer and surrounding tissue showed better prognostic performance than the current staging system [29]. Microsatellite instability, tumor mutational burden, PD-L1 expression, tissue infiltrating lymphocyte, and inflammatory gene expression have been suggested as prognostic and predictive markers of immune check point blockade [30]. Biomarkers from the peripheral blood, however, can be the most useful in clinical utility in terms of easy, relatively safe, and repetitive accessibility. As the peripheral biomarker, specific cells including CD8+ T cell, natural killer cell, myeloid-derived suppressor cell, and regulatory T cell have been investigated [31]. Although the current study could not evaluate these specific cells itself on OS, it may be useful to evaluate the immune and inflammation status using simple complete blood differential counts as peripheral biomarker through further research on the relationship between SIMs and specific cells.

Interestingly, SIMs did not show a significant prognostic performance in ALBI grade III or subgroup treated with curative intent. This finding could be related to the fact that curative intent treatment, including surgical resection, liver transplantation, or ablation, might successfully suppress the effects of tumor aggressiveness or surrounding microenvironment that can be represented by SIMs in a short period of time. Therefore, by using SIMs in addition to a pre-existing staging system, including BCLC, there is the chance to improve treatment outcomes by modifying management, like applying more curative-intent aggressive treatment, including liver transplantation, in HCC patients with high scores of SIMs. To generalize of the current study outcomes and apply to real clinical situations, large-scale external validation should be essential.

The present study had several limitations. Although this was a cohort study based on prospectively obtained data, selection bias may have occurred owing to the large volume, tertiary referral, single-institutional design. Moreover, the primary endpoint of the present study was OS rather than recurrence-free survival, which limits the ability of SIMs on tumor progression. Furthermore, more than 70% of the patients enrolled in the present study had HBV-related HCC, similar to the outcomes of other studies from South Korea [32,33]. 

Despite these limitations, the results of the present study are highly reliable because the study involved prospectively collected data from more than 8700 patients with newly diagnosed, previously untreated HCC from an academic hospital with a large number of highly experienced HCC experts. 

## 4. Materials and Methods

This study was a single-institutional retrospective study based on prospectively collected data from a registry. To identify the prognostic significance and optimal cut-off level of SIMs, data of 6619 consecutive patients from the HCC registry from January 2005 to December 2013 were used. To validate the findings of the training set, data of 2084 consecutive patients from the same registry from January 2014 to December 2016 were used. The details of the HCC registry have been described previously [34]. This study was approved by the Samsung Medical Center Institutional Review Board, and the need for informed consent was waived. Ethical Approval: All procedures performed in studies involving human participants were in accordance with the ethical standards of the institutional and/or national research committee and with the 1964 Helsinki Declaration and its later amendments or comparable ethical standards. The study protocol was approved by the Institutional Review Board of Samsung Medical Center (SMC IRB 2019-04-148).

During the study period, the diagnosis and treatment of HCC were performed according to the “2003, 2009, 2014 practice guidelines for the management of hepatocellular carcinoma” published by the Korean Liver Cancer Association (former Korean Liver Cancer Study Group) and National Cancer Center, Korea [35]. Most of all, however, aggressive intrahepatic tumor control and/or maintenance of liver function status were considered as the most important issues in the management of patients. Briefly, curative treatment including surgical liver resection, RFA, and/or liver transplantation is a priority in both primary and recurrent cases according to the patient performance status, liver function, and tumor location, among other factors. As the next-line modality, TACE was considered, especially for patients with multiple bilateral HCC involvement, and radiotherapy was combined with TACE to maximize local control, mainly for locally advanced HCC with or without vascular tumor thrombosis. Systemic therapy, mainly sorafenib, was recommended for cases that failed to show response to the above-mentioned local modalities or had extensive metastatic lesions. The diagnosis and decision regarding the management for unusual cases were made by a HCC tumor board that was held once a week and consisted of hepatologists, surgeons, diagnostic/intervention radiologists, pathologists, radiation oncologists, and medical oncologists.

### 4.1. Cohorts from the Samsung Medical Center HCC Registry

The HCC registry of the Samsung Medical Center has continued to register and record the baseline clinical/tumor characteristics and primary treatment modality of newly diagnosed, previously untreated HCC since 2005. The following data were collected prospectively for all patients at the time of HCC diagnosis: age, sex, date of diagnosis, etiology of HCC, height, weight, number of tumors, maximum tumor size, presence of vascular invasion, presence of extrahepatic spread, tumor stage (AJCC/UICC TNM, and BCLC stage), ECOG performance status score, laboratory test results (aspartate aminotransferase, alanine aminotransferase, AFP, PIVKA-II, albumin, bilirubin, prothrombin time, activated partial prothromboplastin time, creatinine, and sodium), Child-Pugh score/classification, and initial treatment modality. Moreover, complete blood cell count and differential count were performed in all cases at the time of HCC diagnosis, although data were not included in the HCC registry. 

### 4.2. Laboratoristic Methodology for Blood Count

Complete blood cell count and differential count were performed in all cases at the time of HCC diagnosis, although data were not included in the HCC registry. Two milliliter peripheral venous blood samples were utilized for blood count from each patient at diagnostic work-up for HCC. The blood samples were collected in the ethylenediaminetetraacetic acid anticoagulant vacuum tube and immediately mixed with the anticoagulant after the extraction to prevent the formation of blood clots. The electronic cell counting method was applied using hematology analyzer (XN-9000, Sysmex®, Kobe, Japan). The normal ranges of absolute lymphocyte count, absolute neutrophil count, and platelet count, which constitutes the SIM in our institute, are summarized in Appendix A.

NLR, PLR, and LRM were calculated by dividing the absolute number of neutrophils, platelet, and lymphocytes by the absolute number of lymphocytes, lymphocytes, and monocytes, respectively. 

### 4.3. Training Set 

The primary endpoint of this study was overall survival (OS). To identify the prognostic significance and optimal cut-off level of inflammation markers and to develop a prognostic model, data of 6619 consecutive patients who were treatment naïve and were newly diagnosed with HCC at the Samsung Medical Center from January 2005 to December 2013 and registered in the prospective HCC registry were used.

### 4.4. Validation Set

To validate the new prognostic model and compare the discriminatory abilities between the prognostic models, data of 2084 consecutive patients who were treatment naïve and newly diagnosed with HCC at the Samsung Medical Center from January 2014 to December 2016 and registered in the same HCC registry were used. 

### 4.5. Statistical Analysis

The primary endpoint of this study was OS. OS was estimated using the Kaplan–Meier method and measured from the date of HCC diagnosis to the date of death or last follow-up visit, as assessed on 14 January 2019. The cut-off value of SIMs including NLR, LMR, and PLR was determined based on the point where the C-index of each ratio was maximized using the Cox regression analysis and the values already verified in previous studies [9].

The chi-squared test or Fisher’s exact test for categorical variables and the Mann–Whitney *U* test for continuous variable were used to evaluate the correlation between SIMs and other prognostic factors in the training and validation sets.

The Cox proportional hazards models with the stepwise selection method were used for multivariate analysis. The significant factors of OS that had *p*-values <0.05 on univariate analysis were included in multivariate analysis. The bootstrap method was used to validate the prognostic model in the training set. 

Statistical analysis was performed using SPSS 24.0 software for Windows (SPSS, Chicago, IL, USA), SAS version 9.4 (SAS Institute, Cary, NC, USA), and R 3.4.0 (Vienna, Austria; http://www.R-project.org/). A *p*-value <0.05 was considered statistically significant.

## 5. Conclusions

The SIMs including NLR, PLR, and LMR, showed significant prognostic ability for OS, independent of the liver function and tumor extent, although these factors were significantly correlated with SIMs. Nevertheless, laboratory and/or clinical studies are needed to determine the correlation and mechanism between SIMs and liver function, tumor progression, and/or suppression.

## Figures and Tables

**Figure 1 cancers-12-01300-f001:**
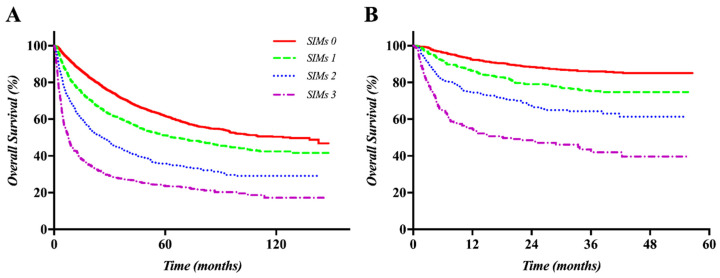
Kaplan–Meier curves of OS according to the prognostic model of systemic inflammation markers (SIMs) in the training and validation sets: A clear difference can be observed in the survival curves according to the score of SIMs in both the training set (**A**) and validation set (**B**).

**Figure 2 cancers-12-01300-f002:**
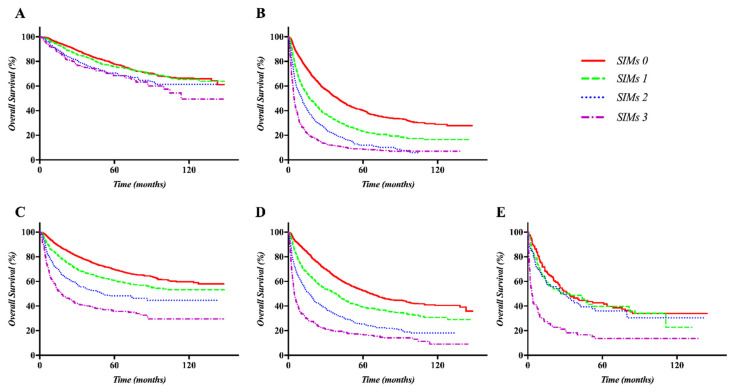
Kaplan–Meier curves of OS according to the SIMs and the treatment policy or ALBI grade in the training set: Although the curves were not significantly separated in the subgroup treated with curative intent (**A**), they were well separated according to the SIMs in only the subgroup of patients treated with palliative aim or supportive care (**B**) or in the subgroup with ALBI grade 1 (**C**) and 2 (**D**). The curves, however, were not significantly separated in the subgroup with ALBI grade 3 (**E**).

**Figure 3 cancers-12-01300-f003:**
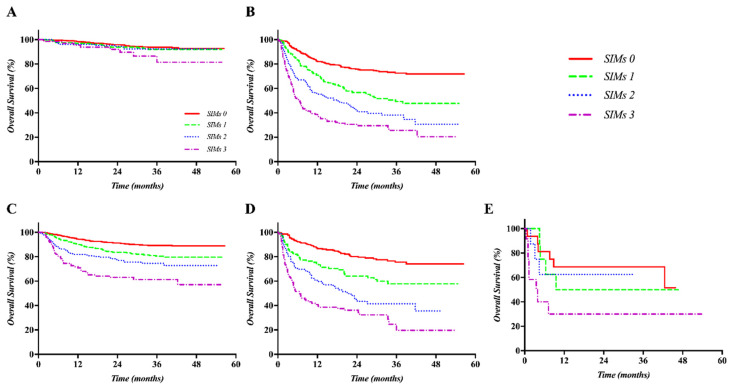
Kaplan–Meier curves of OS according to the SIMs and the treatment policy or ALBI grade in the validation set: Although the curves were not significantly separated in the subgroup treated with curative intent (**A**), they were well separated according to the SIMs in only the subgroup of patients treated with palliative aim or supportive care (**B**) or in the subgroup with ALBI grade 1 (**C**) and 2 (**D**). The curves, however, were not significantly separated in the subgroup with ALBI grade 3 (**E**).

**Table 1 cancers-12-01300-t001:** Baseline Characteristics of the Patients in the Training Set (from 2005 to 2013) and the Validation set (from 2014 to 2016).

Variables	Training Set (*n* = 6619)	Validation Set (*n* = 2084)	*p*-Value
Age (years)			<0.001
Median	57	59	
Range	13–88	21–89	
Sex			0.64
Male	2875 (82.0)	2411 (77.5)	
Female	633 (18.0)	699 (22.5)	
ECOG performance status			<0.001
0	6020 (91.0)	2004 (97.9)	
1	467 (7.1)	41 (8.1)	
2	62 (0.9)	1 (0.0)	
3	46 (0.7)	0 (0.0)	
4	24 (0.4)	1 (0.0)	
Cause of hepatitis			0.002
HBV	4970 (75.1)	1554 (74.6)	
HCV	640 (9.7)	182 (8.7)	
HBV/HCV	59 (0.9)	19 (0.9)	
Alcohol	287 (4.3)	134 (6.4)	
Unknown	663 (10.0)	195 (9.4)	
Child-Turcotte-Pugh class			<0.001
A	5602 (84.6)	1856 (89.1)	
B	895 (13.5)	199 (9.5)	
C	122 (1.8)	29 (1.4)	
BCLC stage			<0.001
0	1020 (15.4)	398 (19.1)	
A	3016 (45.6)	789 (39.7)	
B	761 (11.5)	175 (8.4)	
C	1643 (24.8)	694 (33.3)	
D	179 (2.7)	28 (1.3)	
ALBI grade			<0.001
I	3509 (53.0)	1503 (72.1)	
II	2787 (42.1)	537 (25.8)	
III	323 (4.9)	44 (2.1)	
Portal vein invasion			<0.001
Vp0	5489 (82.9)	1476 (70.9)	
Vp1	431 (6.5)	336 (16.1)	
Vp2	182 (2.7)	1 (0.0)	
Vp3	125 (1.9)	137 (6.6)	
Vp4	392 (5.9)	133 (6.4)	
T stage			0.19
1	1218 (18.4)	366 (18.2)	
2	2916 (44.1)	856 (42.5)	
3	1953 (29.5)	602 (29.9)	
4	532 (8.0)	191 (9.5)	
N stage			0.88
0	6195 (93.6)	1952 (93.7)	
1	424 (6.4)	131 (6.3)	
M stage			<0.001
0	6313 (95.4)	2026 (97.3)	
1	306 (4.6)	57 (2.7)	
AFP (ng/mL)			<0.001
Median	38	20	
Range	1–600,000	1–200,000	
PIVKA-II (mAU/mL)			<0.001
Median	53	77	
Range	2–75,000	6–75,000	
Primary treatment			<0.001
Liver transplantation	130 (2.0)	21 (1.0)	
Hepatectomy	1873 (28.3)	781 (37.5)	
Radiofrequency ablation	1321 (20.0)	350 (16.8)	
TACE	2630 (39.7)	693 (33.3)	
Systemic therapy	255 (3.9)	77 (3.7)	
Radiotherapy	32 (0.5)	37 (1.8)	
None	378 (5.7)	125 (6.0)	
NLR			0.006
Median	1.8	1.88	
Range	0.1–47.8	0.3–46.9	
PLR			<0.001
Median	82.1	88.5	
Range	2.8–793.7	13.0–1491.7	
LMR			0.01
Median	4.0	3.9	
Range	0.3–92.0	0.2–40.0	

Abbreviations: ECOG = Eastern Cooperative Oncology Group; HBV = hepatitis B virus; HCV = hepatitis C virus; BCLC = Barcelona Clinic Liver Cancer; ALBI = albumin-bilirubin; AFP = alpha-fetoprotein; PIVKA-II = Protein induced by vitamin K absence or antagonist-II; TACE = trans-arterial chemo-embolization; NLR = neutrophil-to-lymphocyte ratio; PLR = platelet-to-lymphocyte ratio; LMR = lymphocyte-to-monocyte ratio.

**Table 2 cancers-12-01300-t002:** Univariate Analysis of Overall Survival (OS) According to the Probable Prognostic Factors.

Variables		HR	95% CI	*p*-Value
Age (years)		1.001	0.997–1.004	0.61
Sex	Female	1	-	<0.0001
	Male	1.211	1.108–1.323	
ECOG performance status	0–1	1	-	<0.0001
	2–4	3.229	2.680–3.891	
Etiology	HBV	1	-	0.24
	HCV	0.952	0.846–1.072	
	Alcohol	0.847	0.706–1.017	
	Others	0.953	0.853–1.065	
ALBI grade	I	1	-	<0.0001
	II	1.931	1.797–2.075	
	III	2.392	2.073–2.759	
AFP	<100 ng/mL	1	-	<0.0001
	≥100 ng/mL	2.392	2.232–2.562	
PIVKA-II	<100 IU/mL	1	-	<0.0001
	≥100 IU/mL	3.271	3.042–3.518	
T stage	1	1	-	<0.0001
	2	1.988	1.746–2.264	
	3	5.100	4.484–5.801	
	4	14.488	12.477–16.823	
N stage	0	1	-	<0.0001
	1	3.977	3.556–4.448	
M stage	0	1	-	<0.0001
	1	7.537	6.654–8.537	
Portal vein invasion	Vp0	1	-	<0.0001
	Vp1	4.353	3.887–4.875	
	Vp2	4.862	4.119–5.739	
	Vp3	6.996	5.771–8.481	
	Vp4	8.581	7.653–9.622	
Treatment aim	Curative	1	-	<0.0001
	Palliative	1.248	0.847–1.839	
	None	4.265	2.920–6.230	
NLR	<3.0	1	-	<0.0001
	≥3.0	2.501	2.318–2.698	
PRL	<100.0	1	-	<0.0001
	≥100.0	1.863	1.737–1.997	
LMR	>3.0	1	-	<0.0001
	≤3.0	2.343	2.181–2.517	

Abbreviations: HR = hazard ratio; CI = confidence interval; ECOG = Eastern Cooperative Oncology Group; HBV = hepatitis B virus; HCV = hepatitis C virus; BCLC = Barcelona Clinic Liver Cancer; ALBI = albumin-bilirubin; AFP = alpha-fetoprotein; PIVKA-II = Protein induced by vitamin K absence or antagonist-II; NLR = neutrophil-to-lymphocyte ratio; PLR = platelet-to-lymphocyte ratio; LMR = lymphocyte-to-monocyte ratio.

**Table 3 cancers-12-01300-t003:** Multivariate Analysis of Overall Survival According to the Prognostic Factors.

Variables		HR	95% CI	*p*-Value
Sex	Female	1	-	0.008
	Male	1.138	1.035–1.252	
ECOG performance status	0–1	1	-	0.01
	2–4	1.304	1.062–1.600	
ALBI grade	I	1	-	<0.0001
	II	1.746	1.617–1.884	
	III	1.786	1.518–2.101	
AFP	<100 ng/mL	1	-	<0.0001
	≥100 ng/mL	1.483	1.372–1.602	
PIVKA-II	<100 IU/mL	1	-	<0.0001
	≥100 IU/mL	1.620	1.488–1.764	
T stage	1	1	-	<0.0001
	2	1.375	1.198–1.578	
	3	1.812	1.562–2.103	
	4	2.291	1.883–2.786	
N stage	0	1	-	0.001
	1	1.233	1.088–1.397	
M stage	0	1	-	<0.0001
	1	2.219	1.922–2.562	
Portal vein invasion	Vp0	1	-	<0.0001
	Vp1	1.545	1.344–1.775	
	Vp2	1.661	1.377–2.002	
	Vp3	1.713	1.358–2.161	
	Vp4	2.135	1.841–2.475	
Treatment aim	Curative	1	-	<0.0001
	Palliative	2.696	2.462–2.952	
	None	10.025	8.496–11.829	
NLR	<3.0	1	-	<0.0001
	≥3.0	1.261	1.141–1.393	
PRL	<100.0	1	-	<0.0001
	≥100.0	1.249	1.145–1.361	
LMR	>3.0	1	-	<0.0001
	≤3.0	1.163	1.062–1.273	

Abbreviations: HR = hazard ratio; CI = confidence interval; ECOG = Eastern Cooperative Oncology Group; ALBI = albumin-bilirubin; AFP = alpha-fetoprotein; PIVKA-II = Protein induced by vitamin K absence or antagonist-II; NLR = neutrophil-to-lymphocyte ratio; PLR = platelet-to-lymphocyte ratio; LMR = lymphocyte-to-monocyte ratio.

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
