# Peer review of "Clinical Significance of Systemic Inflammation Markers in Newly Diagnosed, Previously Untreated Hepatocellular Carcinoma"

_cancers, 2020, doi:10.3390/cancers12051300_

Round 1
Reviewer 1 Report
Jeong Il Yu et al describe a clinical significance of systemic inflammation markers (SIMs)—including neutrophil to lymphocyte ratio (NLR), platelet to lymphocyte ratio (PLR), and lymphocyte to monocyte ratio (LMR)—in patients with newly diagnosed previously untreated hepatocellular carcinoma (HCC). they suggest the SIMs they proposed may be better than the current available classification/ranking system. These SIMs are significantly correlated with liver function and tumor extent. More importantly, these SIMs showed the significant prognostic ability for overall survival (OS) in patients with newly diagnosed previously untreated HCC. there are several immune-based cancer diagnosis and prognosis predictive markers been reported, the authors may need to discuss them. Some comments,
- line 41, the authors should check the most updated statistics for HCC/liver cancer.
- line 43, write-out name of TNM should be included.
- In Figures 2 and 3, what is the explanation that no difference of SIMs in the curves of the subgroup treated with curative intent and ALBI grade 3 in both the training and validation sets? In the OS curves, will it be better if take into account of HCC etiology or liver functions? Is there a way to have a better prognosis model?
- Can the authors imply their SIMs into the clinical data of the public available HCC database, such as TCGA to validate their findings?
Author Response
JeongIl Yu et al describe a clinical significance of systemic inflammation markers (SIMs)—including neutrophil to lymphocyte ratio (NLR), platelet to lymphocyte ratio (PLR), and lymphocyte to monocyte ratio (LMR)—in patients with newly diagnosed previously untreated hepatocellular carcinoma (HCC). they suggest the SIMs they proposed may be better than the current available classification/ranking system. These SIMs are significantly correlated with liver function and tumor extent. More importantly, these SIMs showed the significant prognostic ability for overall survival (OS) in patients with newly diagnosed previously untreated HCC. there are several immune-based cancer diagnosis and prognosis predictive markers been reported, the authors may need to discuss them.
→ Thank you for your valuable comments. We inserted following paragraph in discussion section “With the notable achievement in the field of immune-oncology, there is a growing interest and need for valuable biomarkers encompasses tumor microenvironment including immunity. As a prognosticator, the Immunoscore based on the immune cells infiltrate cancer and surrounding tissue showed better prognostic performance than the current staging system [31]. And, microsatellite instability, tumor mutational burden, PD-L1 expression, tissue infiltrating lymphocyte, and inflammatory gene expression have been suggested as prognostic and predictive marker of immune check point blockade [32]. Biomarkers from the peripheral blood, however, can be the most useful in clinical utility in terms of easiness, relative safeness and repetitive accessibility. As the peripheral biomarker, specific cells including CD8+ T cell, natural killer cell, myeloid-derived suppressor cell, and regulatory T cell have been investigated [33]. Although the current study could not evaluate these specific cells itself on OS, it may useful to evaluate the immune and inflammation status using simple complete blood differential counts as peripheral biomarker through further researches on relationship between SIMs and specific cells.”
Some comments,
- line 41, the authors should check the most updated statistics for HCC/liver cancer.
→ Thank you for your thoughtful comment. We changed this sentence as “Liver cancer, HCC accounts for approximately 90% of which, remains the second-leading cause of cancer ranked by absolute years of life lost worldwide [1],”
- line 43, write-out name of TNM should be included.
→ Thank you for your important comment. We inserted following comment in the sentence as “Although the TNM staging system of the American Joint Committee on Cancer (AJCC)/International Union Against Cancer (UICC) provides information about the classification of HCC”
- In Figures 2 and 3, what is the explanation that no difference of SIMs in the curves of the subgroup treated with curative intent and ALBI grade 3 in both the training and validation sets?
→ Thank you for your insightful comment. We determined that the prognostic ability of SIMs could be limited in BCLC 0, or subgroup treated with curative intent, which showed excellent prognosis, because SIMs itself is also an independent prognostic factor, but at the same time it is clearly correlated with tumor burden and liver function. In addition, it could be assumed that curative intent treatment including surgery or ablation could be successfully suppressed the effects of tumor aggressiveness or surrounding microenvironment that can be represented by SIMs in a short period of time. Conversely, the prognostic ability of it could be also limited in BCLC D or ALBI grade 3 for the same reason. There were also very few ALBI grade 3 patients.
We inserted following comments in discussion section as “Interestingly, SIMs did not show a significant prognostic performance in ALBI grade III or subgroup treated with curative intent. This finding could be related that curative intent treatment including surgical resection, liver transplantation or ablation might be successfully suppressed the effects of tumor aggressiveness or surrounding microenvironment that can be represented by SIMs in a short period of time. Therefore, by using SIMs in addition to preexisting staging system including BCLC, there is the chance to improve treatment outcomes by modifying management like applying more curative-intent aggressive treatment including liver transplantation in HCC patients with high scores of SIMs.”
- In the OS curves, will it be better if take into account of HCC etiology or liver functions? Is there a way to have a better prognosis model?
→ Thank you for your insightful comment. As you commented, the prognostic performance is increased as shown in the following figures, when the SIMs and liver function or UICC are combined.
<training set> <validation set>
<training set> <validation set>
We believe that combination of those important factors can develop more accurate prognostic model through further prospective study and external validation.
- Can the authors imply their SIMs into the clinical data of the public available HCC database, such as TCGA to validate their findings?
→ Thank you for your very insightful comment. We also strongly agree that external validation is essential to confirm the clinical significance of SIMs in HCC. Although we tried a lot to do external validation of our results, but it was not easy to get a suitable large-scale cohort. In particular, laboratory results to identify SIMs were not presented in public data such as TCGA. Nevertheless, we strongly agree that a lot of external validation should be required to generalize our results, we will continue to try to do it. We inserted following comments in discussion section “To generalize of the current study outcomes and apply to real clinical situation, large scale external validation should be essential.”

Reviewer 2 Report
- Please, provide figures with higher resolution. Lines are not well distinguishable.
- Please, provide a more complete list of abbreviations and be sure to spell each on the first time that appear. Sometimes is difficult to follow the read due to too much acronyms.
- Is a steatosis index included in your parameters to evaluate patients?
- Please, specify with more details how NLR, PLR, LMR was did by techniques and calculation.
- Which were the criteria used to built up the training set (over OS)?
- Do the authors have data about Glasgow prognostic
score?
Authors present a well organized biostatistic paper describing SIMs and OS in HCC with poor laboratoristic details. More clarifications are needed.
Author Response
- Please, provide figures with higher resolution. Lines are not well distinguishable.
→ Thank you for your important comment. We colored the lines of figures.
- Please, provide a more complete list of abbreviations and be sure to spell each on the first time that appear. Sometimes is difficult to follow the read due to too much acronyms.
→ Thank you for your very insightful comment. We modified the list of abbreviations, and checked them to spell on the first time that appear.
- Is a steatosis index included in your parameters to evaluate patients?
→ Thank you for your important comment. We wanted to evaluate hepatic steatosis index (HSI) in 10,724 Korean (Lee JH, Kim D, Kim HJ et al. Hepatic steatosis index: A simple screening tool reflecting nonalcoholic fatty liver disease Digestive and Liver Disease 42 (2010) 503–508). Unfortunately, however, information on diabetes was only obtained from the period of validation set. HSI was a significant prognostic factor of OS in validation set.
HSI is significantly correlated with LMR. We inserted those results in supplementary tables 5-7 and comments in Result section as follows: “Hepatic steatosis index (HSI) was only available to analyze in the validation set [16], there was a significant correlation between HSI and LMR.”
- Please, specify with more details how NLR, PLR, LMR was did by techniques and calculation.
→ Thank you for your important comments. We inserted following section in Method part.
“4.2. Laboratoristic Methodology for Blood Count
Complete blood cell count and differential count were performed in all cases at the time of HCC diagnosis, although data were not included in the HCC registry. Two milliliter peripheral venous blood samples were utilized for blood count from each patient at diagnostic work-up for HCC. The blood samples were collected in the ethylenediaminetetraacetic acid anticoagulant vacuum tube and immediately mixed with the anticoagulant after the extraction to prevent the formation of blood clot. The electronic cell counting method was applied using hematology analyzer (XN-9000, Sysmex®, Kobe, Japan). The normal ranges of absolute lymphocyte count, absolute neutrophil count, and platelet count which constitutes the SIM in our institute are summarized in the Table S8.
NLR, PLR, and LRM were calculated by dividing the absolute number of neutrophils, platelet, and lymphocytes by the absolute number of lymphocytes, lymphocytes, and monocytes, respectively.”
- Which were the criteria used to built up the training set (over OS)?
→ Thank you for your important comment. Firstly, cut-off value of SIMs including NLR, LMR, and PLR for OS was determined based on the point where the C-index of each ratio was maximized using the Cox regression analysis. And, the maximum C-index of NLR, PLR, and LMR was observed in this study at the point of 2.2 (C-index 0.609, 95% CI 0.600–0.618), 97.5 (C-index 0.589, 95% CI 0.581–0.598), and 3.7 (C-index 0.611, 95% CI 0.602–0.619), respectively. There is a possibility that those cut-off values obtained in this study could be more appropriate in newly diagnosed HCC patients. We decided, however, to use the previously validated cut-off values in many studies, because C-index of these values are not quite inferior with the values obtained in our study as 3.0 for NLR (C-index 0.594, 95% CI 0.586–0.602), 100.0 for PLR (C-index 0.589, 95% CI 0.580–0.597), and 3.0 for LMR (C-index 0.601, 95% CI 0.602–0.619) and the results of our study had not been validated as externally so far. For this point and for further study, we presented all C-index for each value in Supplementary Table 1.
- Do the authors have data about Glasgow prognostic score?
→ Thank you for your good comments. We evaluated the Glasgow prognostic score according to your advice. The CRP test, unfortunately, was not a routine baseline test included in our registry data, so only some patients were possible to analyze. Nevertheless, GPS was a significant prognostic factor of OS both in training and validation sets.
<training set> <validation set>
Furthermore, GPS was also significantly correlated with SIMs in this cohort both in training and validation sets. We inserted this outcome in supplementary tables 2-7 and comments in Result section as follows: “Glasgow Prognostic Score (GPS) can be analyzed only in patients with selectively had C-reactive protein (CRP) measured [15], showed a significant association with all SIMs. “ and “GPS was also only available in patients with selectively had CRP measured, showed a significant association with all SIMs..”
Authors present a well organizedbiostatistic paper describing SIMs and OS in HCC with poor laboratoristic details. More clarifications are needed.
→ Thank you for your insightful comments. Based on your comments, we revised the manuscript as mentioned above. We appreciate your kind review and comments of our study.

Round 2
Reviewer 1 Report
The authors did not completely address the concerns.
- reference 1 format is wrong and the authors did not read to refernce carefully as in the paper, "Liver cancer, hepatocellular carcinoma (HCC) accounts for approximately 90% of which, 42 remains the second-leading cause of cancer ranked by absolute years of life lost worldwide". Actually the liver cancer is the second-leading cause of cancer in men. The author need to check the source data to get an accurate number for overall death rates.
- references format is wrong. additional numbers need to be deleted.
Reviewer 2 Report
Authors provide a strong comment for each point and a more readable version of the figures. In my concern the article is acceptable in the present form.